# PRS-MED: POSITION REASONING SEGMENTATION IN MEDICAL IMAGING

## ABSTRACT

Recent advances in prompt-based medical image segmentation have enabled clinicians to identify tumors using simple input like bounding boxes or text prompts. However, existing methods face challenges when doctors need to interact through natural language or when position reasoning is required, which involves understanding the spatial relationships between anatomical structures and pathologies. We present PRS-Med, a framework that integrates vision-language models with segmentation capabilities to generate both accurate segmentation masks and corresponding spatial reasoning outputs. Additionally, we introduce the Medical Position Reasoning Segmentation (MedPos) dataset, which provides diverse, spatially-grounded question-answer pairs to address the lack of position reasoning data in medical imaging. PRS-Med demonstrates superior performance across six imaging modalities (CT, MRI, X-ray, ultrasound, endoscopy, skin), significantly outperforming state-of-the-art methods in both segmentation accuracy and position reasoning. Our approach enables intuitive doctor-system interaction through natural language, facilitating more efficient diagnoses. Our dataset pipeline, model, and codebase will be released to foster further research in spatially-aware multimodal reasoning for medical applications. (github available after blind review process).

## 1 INTRODUCTION

In the medical field in general and oncology in particular, doctors typically make diagnoses by examining potential tumor locations and types to evaluate tissue conditions. This makes position reasoning and segmentation visualization crucial for supporting early and accurate diagnoses. As medical assistant agents become more common, models like LLaVA-Med Li et al. (2023), Med-MoE Jiang et al. (2024), HuatuoGPT Chen et al. (2024b), and MedVLM-R1 Pan et al. (2025) have been developed to help detect tumors in medical images and provide reasoning about them. While these models show promise, a key challenge remains: doctors often need to identify unknown tumor locations through implicit questions or conversational interactions. Additionally, during diagnosis, doctors need to know the location of a tumor within an image, not just information about the tumor itself. This is why position reasoning segmentation is important. It helps doctors recognize tumors, which leads to more effective diagnosis and treatment. This technology can also help clinics create automated screening systems, reducing manual costs.

In the natural image domain, several works such as LISA Lai et al. (2024), LLM-SEG Wang & Ke (2024a), and SegLLM Wang et al. (2025) have addressed the challenge of reasoning for segmentation, achieving notable success in enhancing object reasoning, identifying object positions through segmentation, and providing simple reasoning about objects. However, these Multimodal-LLMs are not well-trained on medical imaging, making their application to this field difficult. This is due to the complex nature of medical content and the difficulty of boundary learning in medical segmentation, which out-of-domain models struggle with. For position reasoning segmentation, a VLM's vision model needs to be well-trained on medical images to effectively distinguish and localize tumors and anatomies for reasoning.

We present **PRS-Med**, a framework for **P**osition **R**easoning **S**egmentation in medical imaging. This is a unified method that uses a Multimodal-LLM to perform position-reasoning segmentation from simple questions or commands. Our model outputs both a textual description and a segmenta-

tion mask that highlights the tumor location. PRS-Med acts as an intelligent assistant, answering a doctor's questions and visually indicating the position of tumors or anatomical structures in an interpretable way. Our contributions are four folds:

- To address the lack of datasets and evaluation tools for position reasoning in medical imaging, we create and release the Medical Position Reasoning Segmentation (MedPos) dataset pipeline. This pipeline can build a comprehensive position reasoning dataset designed to generate diverse, spatially grounded question-answer pairs in the medical context.
- We present PRS-Med, a position reasoning model that integrates multimodal vision-language learning with a lightweight TinySAM image encoder. It performs spatially-aware tumor segmentation using implicit natural language prompts.
- We are open-sourcing the dataset pipeline, model, and codebase to help the community develop spatially-aware multimodal LLMs in medical imaging.
- We conduct extensive experiments to show the ability of the PRS-Med in position reasoning and understanding with the referring segmentation ability.

The rest of this paper is organized as follows: in Section 2, we briefly review existing methods related to this research. Then, we introduce position reasoning and the segmentation dataset preparation pipeline in Section 3. Afterward, we introduce our proposed model, PRS-Med, in Section 4. Experiment setups are in Section 5. Results of the model assessment is mentioned in Section 6. Finally, we present the conclusion in Section 7.

## 2 RELATED WORK

**Reasoning Image Segmentation:** Recent advancements in reasoning segmentation have begun to integrate high-level reasoning, particularly through the use of the Multimodal-LLMs. Notable works in the domain include LISA Lai et al. (2024), LLM-Seg Wang & Ke (2024b), and SegLLM Wang et al. (2024b), which include a special [SEG] token used to compact segmentation-specific embeddings from the model. However, applying this method to the medical position reasoning is challenging due to the more comprehensive context of the medical-specific vocabulary. To address this limitation, in our design of PRS-Med, we design a unified model from the Multimodal-LLM with the visual features of medical images, enabling effective position-aware segmentation.

**Medical Image Segmentation:** Traditional Medical image segmentation has long relied on fully supervised CNN-based architectures like U-Net Ronneberger et al. (2015) and its variants, such as ResUNet++ Jha et al. (2019), nnu-net Isensee et al. (2018), DoubleUNet Jha et al. (2020a), TransResUNet Tomar et al. (2022), and Swin-UNet Cao et al. (2022). More recently, several promptable segmentations have emerged as a response to the growing demand for interactive and context-aware medical AI. MedSAM Ma et al. (2024a), SAM-Med2D Cheng et al. (2023) adapts the Segment Anything Model (SAM) Kirillov et al. (2023) to medical settings, supporting box- and point-prompted segmentation. However, MedSAM and SAM-Med2D still lack semantic understanding of positional cues within free-form text. In contrast, BiomedParse Zhao et al. (2024b) directly uses text prompts to infer object shapes and positions, learning implicit position priors. Despite its novelty, BiomedParse does not couple segmentation with contextual reasoning, nor does it support implicit or conversational prompts beyond class names. PRS-Med integrates position reasoning with segmentation, enabling an interactive framework that responds to contextual questions and generates both position reasoning outputs and corresponding segmentation masks.

**Multimodal-LLM in Medical Imaging:** Multimodal large language models (MLLMs) have recently shown promising results in medical image reasoning tasks. Prominent methods such as Med-Flamingo Moor et al. (2023), Med-MoE Moor et al. (2023), GSCo He et al. (2024), HuatuoGPT Chen et al. (2024b), and MedVLM-R1 Pan et al. (2025) are built upon vision and Language models like LLaVA Liu et al. (2023b), Qwen2-VL Wang et al. (2024a), and Multimodal Llama Touvron et al. (2023) through the training technique via visual instruction tuning or reinforcement learning methods. Despite their promising results, these models struggle with medical image segmentation, which is a critical task for accurate disease diagnosis. Moreover, they also inherit spatial reasoning limitations from their Multimodal-LLMs, which is observed in the prior works such as SpatialVLM Chen et al. (2024a), Loc-VLM Ranasinghe et al. (2024), and Spatialrgpt Cheng

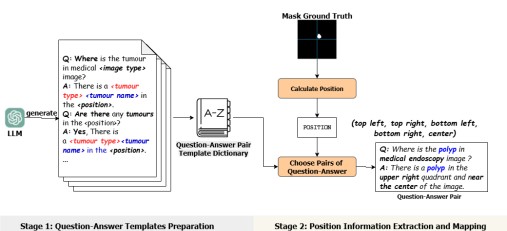

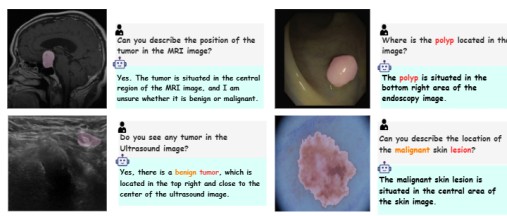

Figure 1: Visualization of two stages of the MedPos dataset pipeline.

Figure 2: Visualization of the segmentation masks and question-answer pairs from the Med-Pos dataset.

et al. (2024). To address these challenges, we propose PRS-Med and the MedPos dataset, which help to extend the Multimodal-LLM capabilities by enabling accurate segmentation and enhancing position reasoning for medical imaging applications.

## 3 MEDPOS DATASET

The research community has proposed several methods for constructing datasets to support multi-modal reasoning, such as LAION Schuhmann et al. (2021), LLaVA Liu et al. (2023b), and Llava-Med Li et al. (2023), which rely on either human annotators or large language models (LLMs) for labeling. However, these approaches face significant challenges when applied to positional reasoning segmentation in the medical domain, due to the limited availability of high-quality datasets with accurate position information along with the tumors or anatomy type. For this reason, we propose the MedPos dataset, which is a Question Answering dataset that includes conversations between the doctor with the assistant to get the position information, type of tumors, or anatomy with related medical information about it.

Our dataset pipeline, as demonstrated in Figure 1, includes two stages: the first stage is the question-answer template preparation, where we prepare the template for the question and answer (which has the supported by doctors for validation the necessary of question and answer template), and the second stage is the Position Information Extraction and mapping, where we do the mapping information about the position, type of tumors/anatomy and related information about the tumor.

**Question-Answer Templates Preparation:** To begin, we leverage the GPT-4 model to generate 50 question-and-answer templates based on the mentioned question-answer pair for training and 5 for testing. These templates are then validated by three doctors to ensure the correctness in the medical context, and to ensure that when combined with the tumor name and positional information, the resulting sentences are coherent and contextually appropriate to provide the necessary information to the doctor.

**Position Information Extraction and Mapping:** We extract positional information from the segmentation mask. Given a binary mask $X_{\mathrm{mask}}$, we first derive the bounding box $\{x, y, w, h\}$, representing the location of tumors within the image. From this, we calculate the center point of the tumor as $x_{\mathrm{center}} = \{x + \frac{w}{2}, y + \frac{h}{2}\}$. Next, we divide the image into four quadrants—top left, top right, bottom left, and bottom right—as illustrated in Figure 1. Based on the location of $x_{\mathrm{center}}$, we determine which quadrant the tumor lies in and assign it a corresponding label. In addition to handling cases where tumors are located near the image center, we also compute the distance between $x_{\mathrm{center}}$ and the geometric center of the image. If this distance falls below a predefined threshold, we label the tumor as being near the center. Finally, we integrate the extracted positional information along with the tumor/anatomy type from the dataset with the question-and-answer templates to generate the final dataset of spatially grounded tumor descriptions. The final samples are demonstrated in Figure 2.

## 4 PRS-MED

**Overall Architecture:** The primary goal of PRS-Med is to perform position reasoning segmentation, enabling the model to explain the location of tumors or anatomies in an image along with relevant medical information. Additionally, the segmentation head allows the model to perform tumor segmentation within the image using a single prompt. The overall architecture is illustrated in Figure 3.

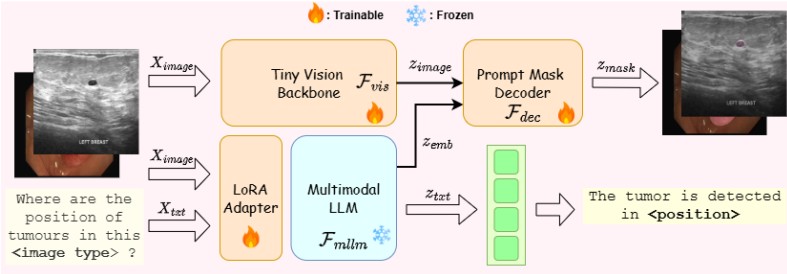

Figure 3: The architecture of PRS-Med comprises three primary components: (1) the Tiny Vision Backbone, (2) the Prompt Mask Decoder, and (3) the Multimodal-LLM. The framework accepts two input modalities: an image and a text-based prompt (e.g., a question). The image is processed through a vision encoder, while the prompt is embedded via a LoRA-adapted Multimodal-LLM. The fused representations are used to produce two outputs: a segmentation mask for the tumor regions, and a textual description specifying the tumor's location.

This framework consists of four main modules. The first is the Vision-Language Model; in our case, we use LLaVA-Med Li et al. (2023), as it is a well-trained Multimodal-LLM for the medical dataset, which we can benefit from its domain of expertise. The second module is the Tiny SAM image encoder, employed from TinySAM Shu et al. (2025), which is used to encode the input image. The third module is the Prompt Mask Decoder, which includes our proposed fusion component that combines image features from the SAM image encoder with the vision-language embeddings from the Medical Vision-Language Model to generate the final segmentation mask. In addition, we include a Language Model Head to perform the reasoning task.

During training, due to the challenges of fine-tuning the full LLaVA-Med model, we apply Low-Rank Adaptation (LoRA) Hu et al. (2022) to enable the model to effectively learn position reasoning information from our prepared dataset.

**Vision Backbone:** The primary objective of the vision backbone is to extract pixel-level features from medical images to support conditional segmentation. For this purpose, we adopt the image encoder from TinySAM Shu et al. (2025), which is based on the lightweight TinyViT architecture Wu et al. (2022). This design enables efficient image encoding while reducing computational resource requirements.

Given a batch of $b$ input images $X_{image} \in \mathbb{R}^{b \times 3 \times W \times H}$, the images are processed through a tiny vision transformer model $\mathcal{F}vis$, consisting of approximately four transformer layers, to produce an image representation embedding $z_{image} \in \mathbb{R}^{b \times 256 \times \frac{W}{16} \times \frac{H}{16}}$. This encoder extracts dense visual features $z_{image}$, which are used for the segmentation task. During training, the encoder is kept unfrozen to allow it to adapt to the medical image domain, thereby improving segmentation performance. The reason for the design choice of this TinyViT-based vision backbone is detailed in Section 5.

By leveraging pre-trained weights from the TinySAM image encoder, our model can better adapt to the medical domain without initializing weights from scratch, which contributes to improved segmentation outcomes.

**Multimodal-LLM**: Most current Multimodal-LLM backbones applied to the medical domain—such as Flamingo Alayrac et al. (2022), LLaVA Liu et al. (2023b), Qwen-VL Wang et al. (2024a), and InternVL Chen et al. (2024c)—demonstrate strong reasoning capabilities. However, they generally lack the ability to generate masks for visual recognition tasks and struggle to comprehend positional information, such as the position of objects within an image. Notably, embeddings from the final layer of these Multimodal-LLMs have proven highly valuable in various applications for semantic understanding, as demonstrated in works like TinyVLA Wen et al. (2025), RoboMamba Liu et al. (2024b), and Groot-N1 Bjorck et al. (2025). Inspired by these insights, we propose a unified design, which leverages semantic embedding from Multimodal-LLM to serve both as a feature extractor for conditioning the masked decoder and as a component for position reasoning. Different from LISA Lai et al. (2024) (the reasoning segmentation approach in the natural image), when they create a new token id for segmentation, in this work, we propose a unified method that leverages directly the joint embedding from the Multimodal-LLM, which can take advantage of the semantics

from the Multimodal-LLM embedding, which can enable the model to understand comprehensive position context in the medical domain.

To generate the Multimodal-LLM embedding $z_{emb} \in \mathbb{R}^{b \times l \times 4096}$ (where $l$ is the token length), and the reasoning output $z_{txt} \in \mathbb{R}^{b \times l}$ from the input image $X_{image} \in \mathbb{R}^{b \times 3 \times w \times h}$ and input text $X_{txt} \in \mathbb{R}^{b \times l \times d}$ (where $d$ is the vocabulary size), we define $F_{mllm}$ as a parametric function implemented using LLaVA-Med Li et al. (2023). The complete process of the model can be described as follows:

$$z_{emb} = \mathcal{F}_{mllm}(X_{image}, X_{txt}), \tag{1}$$

$$z_{txt} = p(z_{txt}|X_{image}, X_{txt}) = \prod_{i=1}^{l} p_\theta(z_{txt}^i|X_{image}, X_{txt}^{i-1}) \text{ with } 1 < i < l. \tag{2}$$

where $\theta$ is the trainable parameter. In our case, $\theta$ is from the parameter of the parametric function $F_{mllm}$, which is the weight of the LLaVA-med model.

During training, due to the high computational cost associated with fully supervised fine-tuning of the LLaVA-Med model, we employ the LoRA method Hu et al. (2022) as an adapter. This approach allows the model to learn reasoning capabilities from our generated position medical reasoning dataset while adapting to generate meaningful embeddings for the mask decoder. Selection and reason for the LoRA hyperparameter are ablation in Section 6.3 and detailed in the Appendix.

**Prompt Mask Decoder**: The goal of this module is to predict the segmented mask from two inputs, including medical images representation feature $z_{image}$ and the embedded image-text prompt $z_{emb}$ from the Multimodal-LLM. This decoder module includes two parts: the fusion module and the mask prediction module. This design allows dynamic alignment between image regions and positional phrases, making better alignment between spatial features and medical vocabulary.

*Fusion Module:* Given the image representation from the vision encoder, denoted as $z_{\text{image}} \in \mathbb{R}^{b \times 256 \times 16 \times 16}$, and the conditioning input from the Multimodal-LLM, denoted as $z_{\text{emb}} \in \mathbb{R}^{b \times l \times 4096}$, the overall fusion process is formalized in Equation 3 and Equation 4.

$$z_{fused} = MHA(\sigma(\frac{\mathcal{F}_{\theta_1}^{proj}(z_{image})\mathcal{F}_{\theta_2}^{proj}(z_{emb})^T}{\sqrt{d_k}})\mathcal{F}_{\theta_2}^{proj}(z_{emb})), \tag{3}$$

$$z_{fused} = z_{fused} + z_{image}. \tag{4}$$

where $d_k$ is the scaling value, $MHA(.)$ is the Multi-head Attention layers, and $\sigma(.)$ is the softmax function.

First, the image representation $z_{\text{image}}$ is reshaped to a new form $z_{\text{image}} \in \mathbb{R}^{b \times (16 \times 16) \times 256}$ to enable interaction with the embedding $z_{\text{emb}} \in \mathbb{R}^{b \times l \times 4096}$ from the Multimodal-LLM. As shown in Equation 3, two projection layers, $\mathcal{F}_{\theta_1}^{\text{proj}}$ and $\mathcal{F}_{\theta_2}^{\text{proj}}$, are applied to project both features into a shared latent space of dimension 256. This alignment allows effective fusion through a cross-attention mechanism, which integrates the image features with the Multimodal-LLM's embeddings. The choice of cross-attention is motivated by the dynamic length of the $z_{\text{emb}}$ sequences, making it a more flexible and suitable alternative to simple addition or concatenation. Following the fusion, a self-attention layer is employed to model the internal dependencies within the target sequence. The resulting fused representation, $z_{\text{fused}} \in \mathbb{R}^{b \times (16 \times 16) \times d}$, is then reshaped to $\mathbb{R}^{b \times 256 \times 16 \times 16}$. Finally, as described in Equation 4, a skip connection is introduced to preserve gradient flow and mitigate the vanishing gradient problem during training.

*Mask Prediction Module:* The input $z_{\text{fused}}$ is passed through a stack of transposed 2D convolutional layers, each followed by Batch Normalization and ReLU activation. This series of operations progressively upsamples $z_{\text{fused}}$ to produce the final segmentation output $z_{\text{mask}} \in \mathbb{R}^{b \times 1 \times 1024 \times 1024}$.

**Objective Function:** This model is trained end-to-end by using the segmentation loss ($\mathcal{L}_{seg}$) and text generation loss $\mathcal{L}_{text}$. The overall objective function is depicted in Equation 5.

$$\mathcal{L} = \lambda_{seg}\mathcal{L}_{seg} + \lambda_{txt}\mathcal{L}_{text}. \tag{5}$$

where $\lambda_{seg}$ and $\lambda_{txt}$ shows the importance of each loss in the overall framework.

Regarding $\mathcal{L}_{seg}$, we employ a combination of Binary Cross-Entropy and Dice loss Sudre et al. (2017), which is a common choice in image segmentation tasks. For $\mathcal{L}_{txt}$, we use the Categorical Cross-Entropy (CE) loss applied on the logit vectors of the tokens output. Let $\hat{y}_{mask}$ denote the

ground truth mask and $z_{mask}$ the predicted mask; similarly, let $\hat{y}_{txt}$ be the ground truth token index sequence and $z_{txt}$ the predicted text logits. Equations 6 and 7 illustrate the formulations of the aforementioned loss functions $\mathcal{L}_{seg}$, and $\mathcal{L}_{txt}$.

$$\mathcal{L}_{seg} = \mathcal{L}_{BCE}(\hat{y}_{mask}, z_{mask}) + \mathcal{L}_{dice}(\hat{y}_{mask}, z_{mask}), \tag{6}$$
$$\mathcal{L}_{text} = \mathcal{L}_{CE}(\hat{y}_{txt}, z_{txt}). \tag{7}$$

By employing this objective function, PRS-Med can simultaneously learn position reasoning while also learn to perform segmentation. Notably, during training, the decoder receives gradients not only from segmentation losses but also from textual reasoning losses, creating a feedback loop where segmentation informs reasoning and vice versa.

## 5 EXPERIMENTAL SETUP

**Dataset:** Our training dataset is constructed by combining several medical data sources images with generated question-answer annotations for 6 different types of images are ultrasound, MRI, RGB image, CT Image, X-ray, and endoscopy images as these are the popular image types, which are mentioned by Biomedparse Zhao et al. (2024a). All of the datapoints are collected BUSI Al-Dhbyani et al. (2020), BrainMRI Cheng et al. (2015; 2016), ISIC Codella et al. (2018), LungCT Konya (2020), LungXray Chowdhury et al. (2020); Konya (2020), Kvasir-SEG Jha et al. (2020b), and ClinicDB Bernal et al. (2015). For the train and test split, we follow the original split from the dataset source to ensure fair comparisons. Furthermore, to increase the difficulty and better evaluate generalization, particularly for polyp tissue segmentation, we augment the test set with additional unseen data from CVC300 Vázquez et al. (2017), ETIS Silva et al. (2014), and ColonDB Tajbakhsh et al. (2016), alongside the test splits from Kvasir-SEG and ClinicDB. This strategy allows for a more rigorous assessment of our method's generalization performance. More detail of the dataset is discussed in the appendix.

**Comparison Baseline:** To compare our work with SOTA methods, we conduct three benchmarks, including segmentation, position reasoning, and position understanding. For the Segmentation task, we compare our methods with the Foundation Segmentation model of medical imaging, such as SAM-Med 2D Zhu et al. (2024) (2024), and Biomedparse Zhao et al. (2024a) (2024) (finetuned image encoder and decoder on our dataset), and the reasoning segmentation model, which is also finetuned on our dataset, is LISA Lai et al. (2024) with two versions are 7B and 13 B. Regarding the SAM model in medical imaging, there is a challenge that most medical segmentation model is based on the box prompt. For this reason, we leverage the Grounding Dino Liu et al. (2024c) as the text understanding model to extract the boxes coordinates for the segmentation task. In the Reasoning and the Position Understanding benchmark, due to the lack of methods done reasoning segmentation, we reproduce the fine-tuning process on our dataset for the Multimodal-LLM for medical image, which includes LLaVA-Med Li et al. (2023) (2024), HuatuoGPT-Vision Chen et al. (2024b) (2024), Med-MoE Jiang et al. (2024), and MedVLM-R1 Pan et al. (2025) (2025) to do the reasoning benchmark. In all of the comparisons, we do the fine-tuning of these methods on our MedPos dataset with the best-practice hyperparameter for each method for the fairest comparison.

**Evaluation Metric:** For the evaluation, we use the mDice, and mIoU to benchmark the segmentation results, as the standard of the medical segmentation task. To assess the fluency in the position reasoning context task, we evaluate through two metrics in the question answering task are ROUGE score and Meteor. In addition, to assess positional understanding accuracy, we use Qwen 2.5 Yang et al. (2024) and Llama 3.1 Grattafiori et al. (2024) to evaluate whether the reasoning position generated by the benchmark models matches the ground truth. This benchmark uses the accuracy metric; a match in positional information is counted as correct (score = 1), while a mismatch is counted as incorrect (score = 0).

## 6 EVALUATION RESULTS

### 6.1 QUANTITATIVE RESULTS

**Segmentation Task Results** To evaluate the overall performance of PRS-Med in the segmentation task, we compare our method with several prior works as aforementioned. Table 1 presents results

on radiology images of six different images and tissues, including Breast Ultrasound, Brain MRI, Lung CT-Scan, Lung X-ray, Polyp Endoscopy and Skin Image.

| Method | Breast Ultrasound | | Brain MRI | | Lung CT-Scan | | Lung X-ray | | Polyp Endoscopy | | Skin Image | |
|---|---|---|---|---|---|---|---|---|---|---|---|---|
| | mDice ↑ | mIoU ↑ | mDice ↑ | mIoU ↑ | mDice ↑ | mIoU ↑ | mDice ↑ | mIoU ↑ | mDice ↑ | mIoU ↑ | mDice ↑ | mIoU ↑ |
| G-Dino + SAM-Med2D Ma et al. (2024b) | 0.515 | 0.441 | 0.667 | 0.625 | 0.540 | 0.392 | 0.401 | 0.300 | 0.488 | 0.418 | 0.237 | 0.171 |
| Biomedparse Zhao et al. (2024a) | 0.783 | 0.698 | 0.294 | 0.245 | 0.516 | 0.399 | 0.972 | 0.949 | 0.824 | 0.774 | 0.893 | 0.822 |
| LISA-7B Lai et al. (2024) | 0.299 | 0.246 | 0.478 | 0.402 | 0.478 | 0.402 | 0.397 | 0.263 | 0.241 | 0.202 | 0.464 | 0.368 |
| LISA-13B Lai et al. (2024) | 0.705 | 0.680 | 0.439 | 0.357 | 0.656 | 0.528 | 0.664 | 0.535 | 0.312 | 0.247 | 0.643 | 0.536 |
| **PRS-Med** | **0.817** | **0.729** | **0.803** | **0.757** | **0.968** | **0.943** | **0.973** | **0.952** | **0.843** | **0.791** | **0.901** | **0.833** |
| *vs previous works* | +0.034 | +0.031 | +0.136 | +0.132 | +0.312 | +0.415 | +0.001 | +0.002 | +0.019 | +0.017 | +0.008 | +0.011 |

Table 1: Quantitative results of PRS-Med across six medical image types. The highest score in each column is in **bold**; the second highest is underlined.

As shown in Table 1, PRS-Med achieves competitive performance compared with the state of the art. Relative to the second-best method, the improvements (mDice, mIoU) are $(+3.4\%, +3.1\%)$ on Breast Ultrasound, $(+13.6\%, +13.2\%)$ on Brain MRI, $(+31.2\%, +41.5\%)$ on Lung CT-Scan, $(+0.1\%, +0.2\%)$ on Lung X-ray, $(+1.9\%, +1.7\%)$ on Polyp Endoscopy, and $(+0.8\%, +1.1\%)$ on Skin Images. These results highlight the generalization and robustness of our method across diverse imaging modalities, anatomical structures, and tumor types.

**Position Reasoning Context Results** To assess the performance of the PRS-Med, we do the evaluation on the position reasoning accuracy with SOTA methods in the Multimodal-LLM for medical images, which is depicted in Table 2.

| Method | Breast Ultrasound | | Brain MRI | | Lung CT-Scan | | Lung X-ray | | Polyp | | Skin Image | |
|---|---|---|---|---|---|---|---|---|---|---|---|---|
| | ROUGE ↑ | METEOR ↑ | ROUGE ↑ | METEOR ↑ | ROUGE ↑ | METEOR ↑ | ROUGE ↑ | METEOR ↑ | ROUGE ↑ | METEOR ↑ | ROUGE ↑ | METEOR ↑ |
| LlaVA-Med Li et al. (2023) | 0.330 | 0.312 | 0.325 | 0.306 | 0.319 | 0.300 | 0.328 | 0.310 | 0.295 | 0.283 | 0.290 | 0.281 |
| HuoGPT Chen et al. (2024b) | 0.363 | 0.459 | 0.355 | 0.440 | 0.348 | 0.431 | 0.360 | 0.446 | 0.301 | 0.322 | 0.298 | 0.310 |
| Med-MoE Jiang et al. (2024) | 0.613 | 0.481 | 0.663 | 0.576 | 0.694 | 0.630 | 0.611 | 0.599 | 0.669 | 0.581 | 0.675 | 0.724 |
| Med-VLMR1 Pan et al. (2025) | 0.281 | 0.289 | 0.276 | 0.284 | 0.270 | 0.280 | 0.278 | 0.285 | 0.250 | 0.263 | 0.242 | 0.259 |
| **PRS-Med** | **0.638** | **0.635** | **0.672** | **0.654** | **0.709** | **0.709** | **0.638** | **0.636** | **0.711** | **0.681** | **0.759** | **0.767** |
| *vs previous works* | +0.025 | +0.154 | +0.009 | +0.078 | +0.015 | +0.079 | +0.027 | +0.037 | +0.042 | +0.100 | +0.084 | +0.043 |

Table 2: Quantitative results of PRS-Med on the reasoning task across six medical image types. The highest score in each column is in **bold**, the second highest is underlined.

As shown in Table 2, PRS-Med attains the highest ROUGE and METEOR on all six datasets. Compared with the strongest prior baseline (Med-MoE), the absolute gains (ROUGE, METEOR) are: Breast Ultrasound $(+0.025, +0.154)$, Brain MRI $(+0.009, +0.078)$, Lung CT-Scan $(+0.015, +0.079)$, Lung X-ray $(+0.027, +0.037)$, Polyp $(+0.042, +0.100)$, and Skin Image $(+0.084, +0.043)$. From these results, we observe that although PRS-Med and LlaVA-Med Li et al. (2023) share the same multimodal-LLM pretraining with the LlaVA-Med, PRS-Med achieves superior performance on the position-reasoning task. The potential reason for this improvement is due to the segmentation module, which is trained jointly with the Multimodal-LLM. This unified training injects consistent localization signals throughout the framework and enables the LoRA adapters to better adapt to the demands of position reasoning by updating their weights accordingly.

**Position Reasoning Accuracy Results:** To evaluate the accuracy of the model in positional reasoning, we adopt a correctness benchmark based on large language models (LLMs). Specifically, Qwen2.5 Yang et al. (2024) and Llama 3.1 Grattafiori et al. (2024) are employed to assess whether the positional information inferred by PRS-Med corresponds to the ground truth. These models are selected as evaluators due to their strong reasoning capabilities and proven accuracy in language comparison. In this evaluation protocol, each correctly inferred position is assigned a score of 1, whereas incorrect inferences receive a score of 0.

| Method | Qwen Benchmark | Llama 3.1 Benchmark | Final result |
|---|---|---|---|
| LLaVA-Med Li et al. (2023) | 42.6% (± 0.6) | 41.2% (± 0.8) | 41.9% |
| HuatuoGPT-Vision Chen et al. (2024b) | 54.2% (± 0.1) | 42.4% (± 0.3) | 48.3% |
| Med-MoE Jiang et al. (2024) | 70.4% (± 0.1) | 58.5% (± 0.1) | 64.5% |
| MedVLM-R1 Pan et al. (2025) | 61.7% (± 0.3) | 49.8% (± 0.7) | 55.8% |
| **PRS-Med** | **75.4% (± 0.1)** | **58.9% (± 0.2)** | **67.2%** |
| *vs previous works* | +5.0% | +0.4% | +2.7% |

Table 3: Qualitative results of **PRS-Med** on position reasoning with SOTA multimodal LLMs. The highest score in each column is in **bold**, the second highest is underlined.

As shown in Table 3, PRS-Med achieves an overall improvement of $+2.7\%$ compared to the second-best method. These competitive results demonstrate that PRS-Med is capable of accurately understanding positional information. This demonstrates that segmentation-informed reasoning improves positional grounding, reducing hallucination in text-only reasoning models.

## 6.2 QUALITATIVE ANALYSIS

**Comparison Visualization:** In Figure 4, we present qualitative visualizations that highlight the improvements achieved by PRS-Med. The results clearly show that PRS-Med can capture small lesions and anatomies that previous baselines miss, consistently generating completed masks with the lowest loss. We attribute these improvements to the informative feature extraction of the lightweight vision encoder and the effectiveness of the fusion module. Overall, the results provide strong evidence for the promise of our approach. In addition, this visualization also highlights the ability of the Multimodal-LLM, which can deal with both the reasoning task and the medical image segmentation task with high accuracy. However, through this visualization, we can observe that the boundary problems are still the limitations of PRS-Med, and we are planning to improve in the future.

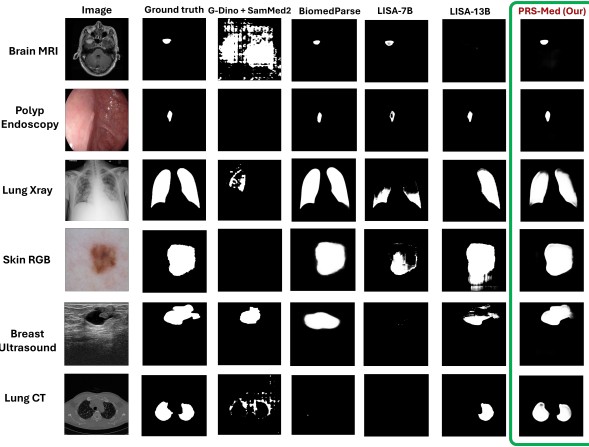

Figure 4: Comparison of PRS-Med with baseline models on polyp detection. PRS-Med produces more accurate boundaries and captures small lesions missed by other methods.

**Reasoning And Segmentation Results On Unseen Data:** Figure 5 shows the reasoning results along with the segmentation in the unseen question templates and in the unseen medical images. In all of the samples, PRS-Med can handle and answer these unseen questions quite effectively although these questions templates are not included in the training dataset. This indicates the position understanding and effectiveness of PRS-Med in the diverse context of language.

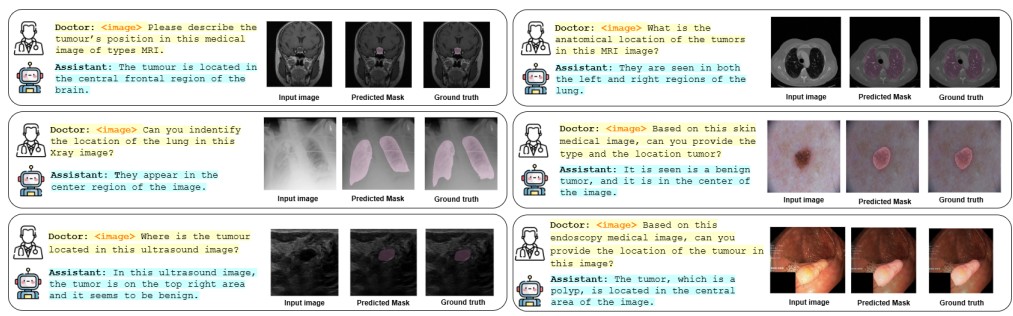

Figure 5: Example of answers, and segmented masks from unseen data of PRS-Med.

## 6.3 ABLATION STUDY

To assess the choice and the effectiveness of the module in our framework, we conduct several experiments to assess the performance and the limitations of each module. The experiments are conducted in the same training and testing dataset with the benchmark. Regarding the metrics, we calculate the average mDice and average mIoU for the segmentation results, and ROUGE, METEOR

for reasoning results on different modalities in our test dataset to have the best assessment of the robustness of each choice.

| Vision Backbone | Parameters | mDice ↑ | mIoU ↑ |
|---|---|---|---|
| SAM-Med (Frozen) | 21.52M | 0.798 | 0.719 |
| SAM-Med (Full) | 292.60M | 0.891 | 0.838 |
| SAM-Med (LoRA) | 47.84M | 0.790 | 0.711 |
| TinySAM (no pretrained) | 31.49M | 0.674 | 0.582 |
| TinySAM (frozen) | 21.73M | 0.737 | 0.662 |
| **TinySAM (Full)** | 31.49M | **0.884** | **0.834** |

| MLLM | mDice ↑ | mIoU ↑ | ROUGE ↑ | METEOR ↑ |
|---|---|---|---|---|
| q,v | 0.573 | 0.483 | 0.478 | 0.436 |
| q, k, v | 0.714 | 0.621 | 0.585 | 0.578 |
| **q,k,v,o** | **0.879** | **0.827** | **0.654** | **0.599** |

Table 4: **(Left):** Comparison results of different vision encoder backbones. **(Right):** Ablation study on LoRA target module (r=16) on Multimodal-LLM.

**Design Choice Of Vision Encoder Backbone:** As described in Section 4, we do the experiment to emphasize our vision encoder choice with the results presented in Table 4 (Left). In our design, we consider two published pre-trained models, SAM-Med Ye et al. (2023) and TinySAM Shu et al. (2025), as our vision encoder. As demonstrated in Table 4, the TinySAM gets higher performance than SAM-Med (LoRA) with similar trainable parameters in the overall framework, and get lower results with the full supervised training version of SAM-Med. However, due to our target of an efficient model, based on the trade-off between performance and the number of parameters, we choose TinySAM as our vision encoder.

**Initialization Of TinySAM Image Encoder:** We observe that the initialization of TinySAM significantly affects the overall results. For this reason, in Table 4 and experiment 5 and 6, we assess the contribution of the TinySAM pretrained weight. Without the pretrained initialization, the overall results drop substantially, which shows the importance of the pretrained initialization to the overall framework.

**Design Choice Of MLLM Backbone** To evaluate the choice of MLLM backbone for PRS-Med, we conducted experiments comparing three models are LLaVA-1.5 Liu et al. (2024a; 2023a), LLaVA-1.6 Liu et al. (2024a; 2023a), and LLaVA-Med—using the same 7B backbone and fine-tuned via the LoRA approach. The comparison focuses on two tasks: segmentation and position reasoning, as shown in Table 5. The results indicate that the overall performance of the LLaVA-Med baseline surpasses that of LLaVA-1.5 and LLaVA-1.6. This improvement can be attributed to LLaVA-Med's enhanced adaptation to the medical domain, which enables it to better handle tasks involving medical data.

| MLLM | Parameters | Avg-mDice ↑ | Avg-mIoU ↑ | ROUGE ↑ | METEOR ↑ |
|---|---|---|---|---|---|
| LLaVA-1.5 (LoRA) | 34.63M | 0.709 | 0.642 | 0.414 | 0.385 |
| LLaVA-1.6 (LoRA) | 31.49M | 0.744 | 0.671 | 0.508 | 0.432 |
| **LLaVA-Med (LoRA)** | 31.49M | **0.879** | **0.827** | **0.654** | **0.599** |

Table 5: Ablation study for design choice of the Multimodal-LLM.

**LoRA Target Modules Choice:** To assess the contribution of the target module from LoRA in the overall framework, we do the ablation to evaluate our choice of target module from LoRA, which is mentioned in Table 4 (Right). Our choice for the target modules (q,k,v,o) makes the overall framework achieve significant higher performances, which indicates that all of the projection weights allow LoRA to more effectively align cross-modal representations for both the reasoning and segmentation tasks.

# 7 CONCLUSION

In conclusion, we introduced PRS-Med, a novel framework that uses natural language prompts to perform spatially-aware tumor segmentation and position-based reasoning in medical images. Our approach integrates a lightweight image encoder with a vision-language model, enabling intuitive, conversational interaction for medical analysis. To address the critical lack of positional reasoning data, we also created and released the MedPos dataset. This dataset combines positional question-answer pairs created from segmentation masks with tumor and anatomical annotations. Our comprehensive evaluation across six different imaging modalities and with state-of-the-art methods shows that PRS-Med is effective in both segmentation and positioning tasks, and it is robust to unseen data. By open-sourcing our dataset, pipeline, model, and codebase, we aim to accelerate research in spatially-aware, multimodal reasoning for medical applications. PRS-Med has the potential to enhance clinical workflows by improving diagnostic accuracy, reducing interpretation time, and enabling a more intuitive interaction between physicians and AI systems.

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

# A  APPENDIX

## A.1  IMPLEMENTATION DETAILS

In our implementation, experiments were conducted using two H100 80GB GPUs. The model was trained for 20 epochs, requiring approximately 24 GPU hours with a batch size of 8 for each GPU. We employed the AdamW Loshchilov & Hutter (2017) optimizer with a learning rate of $1 \times 10^{-4}$, and the best checkpoint was selected at epoch 18 based on validation performance. For the LoRA hyperparameters, we set the rank to 16, used the same value for alpha, and applied a dropout rate of 0.05. The LoRA weight is initialized following a uniform distribution.

## A.2  BROADER IMPACT

The broader impact of PRS-Med lies in its capability for position reasoning segmentation. As an intelligent assistant, it can support doctors in rapid screening and efficiently gather detailed information about a patient's disease status. This, in turn, helps reduce diagnosis and treatment time, enabling more patients to receive care on the same day.

## A.3  MORE DETAILS FOR EVALUATION BY LARGE LANGUAGE MODEL

In this section, we describe the benchmark progress via two LLMs are Qwen 2.5 Yang et al. (2024) and Llama 3.1 Grattafiori et al. (2024). Each model will run three times on these different questions, then for each model, we calculate the average and the standard deviation to get the benchmark results for each agent. Then the final result is calculated by the results of the agents, which means that with more agents, we can reduce the bias of the benchmark from the LLM.

As the LLM also has hallucinations, it can affect to the benchmark results. However, as our observation, the hallucination is minor, which create a tiny effect, not affecting considerably the judging process for the overall performance of the models.

The following list shows the different templates of the benchmark prompts:

1. `As a medical image specialist`

   **Instruction:** Answer the question related to the position content, return only yes or no

   **Prompt:** Given the following question and answer with the ground truth, is the position in the answer similar or same with the ground truth and match with the question. Sample - Question: $\{question\}$ — groundtruth: $\{groundtruth\}$ — Prediction: $\{answer\}$ Return yes if they are similar. Return no if they are different.

2. `As a doctor`

   **Instruction:** Answer the question related to the position content, return only yes or no

   **Prompt:** Check if the location information provided in the prediction aligns with the position mentioned in the ground truth and is relevant to the question. Input — Question: $question$ — Ground Truth: $groundtruth$ — Prediction: $answer$ Respond with Yes if the positions are similar. Respond with No if they are different.

3. `As you are a doctor and you are looking to the medical image:`

   **Instruction:** Answer the question related to the position content, return only yes or no

   **Prompt:** Evaluate whether the predicted answer captures the same or similar positional context as the ground truth, based on the provided question. Question: $question$ Groundtruth: $groundtruth$ Prediction: $answer$ Answer with "Yes" if the position is similar, otherwise "No"

For each "yes" response, it is calculated as one correct answer; for each "no" response, it is calculated as one incorrect answer. The accuracy is calculated by the sum of the correct answers over all of the samples in the test set.

## A.4 Explanation for the Position Identification

In Section 3, we divide the image into 4 quadrants in order to identify the position of tumors in the image, as the representation in Figure 6.

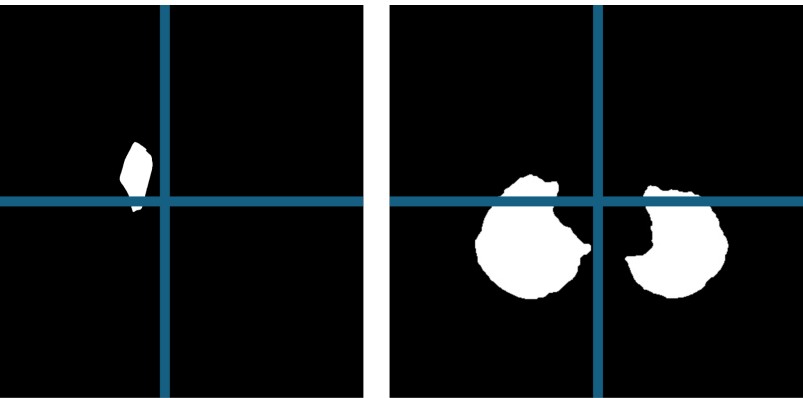

Figure 6: Examples of the position identification for dataset creation.

For images containing a single tumor, we identify the tumor's position by selecting the region that consists of the largest mask area. In the case of images with two tumors, we detect both regions and combine their positional descriptions into a single sentence. Additionally, we compute the Euclidean distance between the center of the image and the centroid of the mask to determine whether a tumor is located near the center. In our pipeline, we set the distance threshold for being considered "near the center" at 20 unit.

For example, in the first image shown in Figure 6, the tumor is located in the top-left region and near the center, while the second image contains two tumors identified in the bottom-left and bottom-right regions, which both tumors are also near the center. The final output sentence describing the tumor positions is as follows:

- There is a tumor in the top left region and near the center.

- A tumor is located in the bottom left quadrant, near the center, and another(or in other words, in our prompt: "," and, with) tumor is located in the bottom right quadrant, near the center.

By using this logic-based approach, we can infer the position of the tumor and map it to the corresponding position vocabulary in our dictionary. This mapped position can then be integrated into a predefined template to generate the final question-answer pairs. The advantage of this method is that it eliminates the need for manual labeling of question-answer pairs, thereby reducing annotation costs and minimizing human error.

## A.5 More Details For Dataset Information

To create the MedPos dataset, we employ several different datasets from the open-source datasets, including BUSI Al-Dhabyani et al. (2020) dataset, ISIC Codella et al. (2018); DiSanto (2023), Kvasir-SEG Jha et al. (2020b), ClinicDB Bernal et al. (2015), ColonDB Tajbakhsh et al. (2016), CVC300 Vázquez et al. (2017), ETIS-Polyric Silva et al. (2014), LungCT Konya (2020), Lung Xray Chowdhury et al. (2020); Rahman et al. (2021), and Brain MRI dataset Cheng et al. (2015; 2016). These datasets focus on six different types of images are ultrasound image, RGB image, endoscopy image, CT-scan image, Lung Xray image, and MRI image. These are the common medical image types in the real-world application, mentioned in BiomedParse Zhao et al. (2024a). In total, we have **28650** images and question-answer pairs for training, and **4647** images along with the question-answer pairs for testing. Due to the limitations of the number of medical images, this dataset is reasonable for the experiments in our paper, and the results show that our methods can adapt well in this dataset setting.

| Image and specification type | Number of train sample | Number of test sample |
|---|---|---|
| Breast Ultrasound | 599 | 113 |
| Skin RBG Image | 900 | 379 |
| Polyp Endoscopy | 1450 | 798 |
| Lung CT | 7959 | 1800 |
| Lung Xray | 16280 | 957 |
| Brain MRI | 1462 | 600 |

Table 6: Statistics for the number of training and test samples for each specification and type of images.

Regarding the number of question-answer pairs, **50 templates** are used for the training, and **5 unseen templates** for the testing.

A.6   QUESTION-ANSWER DATASET CREATION

**Prompt to generate the question-answer pair:** For the generation of the question-answer pair, we leverage the Role Prompting technique. The prompt for the generation is mentioned as following.

1. `As a doctor, you have to work with medical images everyday`

2. **Instruction:** Given the Question and answer template as an example:
   **Q**: Where is the tumor position in the <image type> medical image? **A**: There is a <tumor type> <tumor name> in the <position>.

3. **Prompt:** Generate the informative question and answer pair that keep the template of the provided template.

**Template for question and answer pairs generated:**   For the training dataset, we generate 50 question-answer pair templates, as it is diverse for the language understanding. All of the training questions and answers are mentioned as following:

1. **Q:** In the provided {image_type} scan, where can the tumor be observed?
   **A:** The {tumor_type} tumor can be observed in the {position_description} area of the image.

2. **Q:** Specify the tumor's location in this {image_type} view.
   **A:** The {tumor_type} tumor is clearly seen in the {position_description} region.

3. **Q:** Can you detect where the tumor is in this {image_type}?
   **A:** The {tumor_type} tumor is present in the {position_description} area.

4. **Q:** Which area of the {image_type} shows a tumor?
   **A:** The {tumor_type} tumor is seen in the {position_description}.

5. **Q:** From this {image_type} scan, what is the tumor's location?
   **A:** The {tumor_type} tumor is located in the {position_description} part.

6. **Q:** Could you point out where the tumor is located in this {image_type} scan?
   **A:** The {tumor_type} tumor can be observed in the {position_description} area of the {image_type} scan.

7. **Q:** What part of the body does the tumor appear in this {image_type} medical image?
   **A:** In this {image_type} image, the {tumor_type} tumor is found in the {position_description} section.

8. **Q:** Identify the region in this {image_type} image that shows the tumor.
   **A:** The region showing the {tumor_type} tumor in this {image_type} image is the {position_description}.

9. **Q:** Can you specify the tumor's location based on the {image_type} image provided?
   **A:** Based on the provided {image_type} image, the {tumor_type} tumor lies in the {position_description} region.

10. **Q:** From this {image_type} image, where would you say the tumor is located?
   **A:** Judging from the {image_type} image, the {tumor_type} tumor is located at the {position_description}.

11. **Q:** Could you specify the tumor's location in this {image_type} scan?
   **A:** The {tumor_type} tumor is clearly located in the {position_description} area.

12. **Q:** What region of this {image_type} shows the presence of a tumor?
   **A:** The {tumor_type} tumor is seen in the {position_description} region.

13. **Q:** Where in the anatomical image ({image_type}) is the tumor located?
   **A:** The {tumor_type} tumor is present in the {position_description} portion of the anatomy.

14. **Q:** Is there a visible tumor in the {image_type} image, and where?
   **A:** Yes, the {tumor_type} tumor is found in the {position_description} area.

15. **Q:** In which region of the {image_type} can the tumor be found?
   **A:** The {tumor_type} tumor can be found in the {position_description}.

16. **Q:** Pinpoint the tumor location in this {image_type} scan.
   **A:** The {tumor_type} tumor is located in the {position_description} region.

17. **Q:** Which part of the {image_type} image contains the tumor?
   **A:** The {tumor_type} tumor is contained in the {position_description} part.

18. **Q:** In this {image_type} image, what is the tumor's anatomical position?
   **A:** The anatomical position of the {tumor_type} tumor is {position_description}.

19. **Q:** Identify the segment of this {image_type} that has a tumor.
   **A:** The segment showing the {tumor_type} tumor is {position_description}.

20. **Q:** Where is the abnormal mass located in this {image_type} scan?
   **A:** The abnormal {tumor_type} mass appears in the {position_description}.

21. **Q:** Can you detect the tumor's placement in the {image_type} image?
   **A:** The placement of the {tumor_type} tumor is in the {position_description} zone.

22. **Q:** Is the tumor visible in this {image_type}, and where is it found?
   **A:** Yes, the {tumor_type} tumor is located in the {position_description} portion.

23. **Q:** Which anatomical zone in the {image_type} image shows a tumor?
   **A:** The {tumor_type} tumor is visible in the {position_description} region.

24. **Q:** Where does the tumor appear in this {image_type} scan?
   **A:** The {tumor_type} tumor appears in the {position_description} region of the scan.

25. **Q:** Indicate the region where the tumor is located in this {image_type}.
   **A:** The region of the {tumor_type} tumor is the {position_description}.

26. **Q:** In this scan of {image_type}, where do you see the tumor?
   **A:** The {tumor_type} tumor is seen in the {position_description} area.

27. **Q:** What area in the {image_type} image reveals the tumor?
   **A:** The area showing the {tumor_type} tumor is {position_description}.

28. **Q:** According to this {image_type} image, where is the tumor found?
   **A:** The {tumor_type} tumor is found in the {position_description}.

29. **Q:** What is the approximate tumor position in this {image_type}?
   **A:** Approximately, the {tumor_type} tumor lies in the {position_description}.

30. **Q:** Give the precise tumor location in this {image_type} image.
   **A:** The {tumor_type} tumor is precisely located in the {position_description}.

31. **Q:** Can the tumor be located in the upper or lower part of the {image_type}?
   **A:** The {tumor_type} tumor is found in the {position_description} section.

32. **Q:** Which side of the {image_type} contains the tumor?
    **A:** The {tumor_type} tumor is on the {position_description} side.

33. **Q:** In this {image_type} scan, which quadrant has the tumor?
    **A:** The {position_description} quadrant contains the {tumor_type} tumor.

34. **Q:** What part of the {image_type} is affected by the tumor?
    **A:** The {position_description} part is affected by the {tumor_type} tumor.

35. **Q:** Where is the main tumor mass observed in this {image_type}?
    **A:** The main {tumor_type} tumor mass is observed in the {position_description} region.

36. **Q:** Describe the tumor's spatial location in this {image_type} scan.
    **A:** The spatial location of the {tumor_type} tumor corresponds to the {position_description}.

37. **Q:** Where is the suspicious mass situated in this {image_type}?
    **A:** The suspicious {tumor_type} mass is situated at the {position_description}.

38. **Q:** Which image region shows the most tumor density in this {image_type}?
    **A:** The region with most density of the {tumor_type} tumor is the {position_description}.

39. **Q:** Can you tell which section of the image highlights the tumor?
    **A:** The highlighted {tumor_type} tumor appears in the {position_description} section.

40. **Q:** In this {image_type} medical scan, where can the tumor be localized?
    **A:** The {tumor_type} tumor can be localized in the {position_description} area.

41. **Q:** Where is the focal point of the tumor in this {image_type}?
    **A:** The focal point of the {tumor_type} tumor is at the {position_description}.

42. **Q:** Which directional area of the {image_type} shows the tumor?
    **A:** The {tumor_type} tumor shows up in the {position_description} direction.

43. **Q:** Can you indicate the approximate region where the tumor lies?
    **A:** The {tumor_type} tumor lies approximately in the {position_description}.

44. **Q:** Where would you mark the tumor in this {image_type}?
    **A:** I would mark the {tumor_type} tumor in the {position_description}.

45. **Q:** In this view of the {image_type}, what part contains the tumor?
    **A:** The {tumor_type} tumor is in the {position_description} view.

46. **Q:** What's the visible tumor location in the {image_type} image?
    **A:** Visibly, the {tumor_type} tumor is in the {position_description}.

47. **Q:** According to the image, where does the tumor appear?
    **A:** The {tumor_type} tumor appears in the {position_description} area.

48. **Q:** From the given {image_type}, where can we see the tumor?
    **A:** The {tumor_type} tumor is seen in the {position_description} region.

49. **Q:** What is the rough location of the tumor in the image?
    **A:** Roughly, the {tumor_type} tumor is at the {position_description}.

50. **Q:** Could you highlight the tumor's location in this {image_type} image?
    **A:** The {tumor_type} tumor is highlighted in the {position_description} region.

For the test dataset, we use 5 following templates for testing the performance of the model:

1. **Q:** Can you identify the location of the {tumor_type} tumor in this {image_type} medical image?
   **A:** The {tumor_type} tumor is located in the {position_description} region of the {image_type} image.

2. **Q:** Please describe the {tumor_type} tumor's position in this medical image of type {image_type}.
   **A:** In this {image_type} medical image, the {tumor_type} tumor appears in the {position_description}.

3. **Q:** What is the anatomical location of the {tumor_type} tumor in this {image_type} medical image?
   **A:** The anatomical position of the {tumor_type} tumor in this {image_type} image is {position_description}.

4. **Q:** Based on this {image_type} medical image, can you provide the location of the {tumor_type} tumor in this image?
   **A:** From the {image_type} image, the {tumor_type} tumor is seen in the {position_description} area.

5. **Q:** Where is the {tumor_type} tumor located in this {image_type} medical image?
   **A:** The {tumor_type} tumor is situated in the {position_description} part of the {image_type} image.

