# OpenReview forum: "PRS-MED: POSITION REASONING SEGMENTATION IN MEDICAL IMAGING"
_ICLR.cc/2026/Conference — ICLR 2026 Conference Withdrawn Submission_

### Official Review · Reviewer_2M8a · 2025-10-27

**Soundness:** 2
**Presentation:** 3
**Contribution:** 2
**Rating:** 2
**Confidence:** 4

**Summary:**

This paper proposes PRS-Med, a multimodal model that performs position reasoning segmentation in medical images. Given a medical image and a natural-language query, PRS-Med outputs both an explanatory text answer and a segmentation mask highlighting the tumor’s location. To enable and evaluate this capability, the authors introduce a new dataset called MedPos, which consists of diverse question-answer pairs grounded in six different medical imaging modalities. In experiments, PRS-Med outperforms prior methods: it achieves higher segmentation accuracy and better spatial reasoning scores.

**Strengths:**

1. PRS-Med presents a unified model that combines language understanding with segmentation in a novel way for the medical domain.
2. The authors curated a diverse set of medical images across six modalities and generated associated Q&A pairs that require spatial reasoning.
3. The experimental evaluation is comprehensive and generally demonstrates clear performance gains of PRS-Med.

**Weaknesses:**

1. The novelty of the paper is limited since the idea of such a framework has been explored in prior works like LISA, LLM-Seg, and SegLLM. The contribution is adapting known techniques to a new domain (medical) and task.
2. The MedPos dataset is built synthetically using templated Q&A and automatically extracted positions. This raises concerns about the *depth* and *variety* of spatial reasoning captured. All questions seem to revolve around locating a single tumor in quadrants of the image (e.g. “top-left area”).  It does not reflect the full complexity of clinical spatial reasoning. Moreover, the data distribution is not balanced, where the Lung X-ray dominates the training set.
3. There is a potential concern about the fairness of the comparisons made in the experiments. PRS-Med is specifically trained (via instruction tuning with LoRA) on the new MedPos dataset for both segmentation and QA, whereas it’s not clearly stated if the baseline models were similarly adapted to this dataset. Many baselines (LISA, LLaVA-Med, Med-MoE, etc.) were originally developed for related tasks, but not exactly the same spatial QA with mask task.
4. The writing quality should be improved. The details of the dataset construction are somewhat confusing. For example,  it mentions *“the template... which has the supported by doctors for validation of the necessary question and answer template”*, which is not clearly worded. The term “qualitative results” is used for Table 3 despite showing quantitative improvements, which is a bit confusing (perhaps they meant “additional results”).

**Questions:**

see Weakness

**Details Of Ethics Concerns:**

No Ethics Concerns.

---

### Official Review · Reviewer_wyPy · 2025-10-31

**Soundness:** 2
**Presentation:** 3
**Contribution:** 3
**Rating:** 4
**Confidence:** 3

**Summary:**

The paper introduces PRS-Med, a novel framework designed to jointly perform segmentation and understand natural language prompts related to spatial positioning in medical images, addressing the limitations of specialized VLMs and segmentation models. To solve this, the authors make two main contributions: the MedPos dataset and the PRS-Med model. The dataset is built with a scalable pipeline that automatically generates spatially grounded question-answer pairs by first using GPT-4 to create doctor-validated Q&A templates and then extracting positional information (e.g., "top-left," "near the center") by calculating the centroid of ground-truth segmentation masks. The PRS-Med model is a unified architecture that takes an image and text prompt to output both a textual answer and a precise segmentation mask; it features a trainable TinySAM vision encoder for visual features and a frozen LLaVA-Med model adapted with LoRA for semantic embeddings. The authors demonstrate that this end-to-end trained model, which uses a combined loss for text generation and segmentation, achieves state-of-the-art performance on both segmentation and reasoning tasks across six medical imaging modalities.

**Strengths:**

1. The paper tackles a significant and practical gap. A clinician's workflow is inherently interactive. The ability to ask, "Where is the lesion?" and "Segment the mass in the upper right quadrant" is a far more natural and useful interaction than manually drawing bounding boxes.
2. The MedPos dataset pipeline is a key contribution. By programmatically mapping mask centroids to spatial labels and combining them with doctor-validated templates, the authors have created a scalable way to generate a large-scale positional reasoning dataset. This cleverly bypasses the enormous cost of manual annotation.
3. The choice of a lightweight TinySAM encoder, which is then fully fine-tuned, strikes a satisfactory balance between efficiency and domain adaptation. The ablations show this is a satisfactory trade-off, achieving performance close to a massive SAM-Med model with far fewer parameters.
4. The experimental validation is a major strength.

**Weaknesses:**

1. The current positional labels are coarse. Have you considered extending the MedPos pipeline to generate relative spatial relationships (e.g., "lesion A is adjacent to lesion B" or "tumor near the right lung")? How do you think the model would perform?
2. Given the acknowledged limitations in boundary sharpness, do you attribute the issue to the lightweight TinySAM encoder or the fusion module? Would a better decoder that combines different scale features from the vision backbone after the semantic fusion help improve these edges?
3. Your design fine-tunes the entire vision backbone but only uses a LoRA adapter for the MLLM. What was the rationale for this?
4. The qualitative results show successes. What are the primary failure cases?

**Questions:**

1. The current positional labels are coarse. Have you considered extending the MedPos pipeline to generate relative spatial relationships (e.g., "lesion A is adjacent to lesion B," or "tumor near the right lung")? How do you think the model would perform?
2. Given the acknowledged limitations in boundary sharpness, do you attribute this to the lightweight TinySAM encoder or the fusion module? Would a better decoder that combines different scale features from the vision backbone after the semantic fusion help improve these edges?
3. Your design fine-tunes the entire vision backbone but only uses a LoRA adapter for the MLLM. What was the rationale for this?
4. The qualitative results show successes. What are the primary failure cases?

---

### Official Review · Reviewer_7h8f · 2025-11-01

**Soundness:** 2
**Presentation:** 2
**Contribution:** 2
**Rating:** 2
**Confidence:** 3

**Summary:**

The paper proposes a multimodal framework - PRS-Med, that links medical image segmentation with natural language reasoning. It enables the model to explain spatial relationships in diagnostic images. Besides, the paper introduces MedPos, a dataset of spatially grounded QA pairs across various imaging modalities, and integrates a lightweight vision encoder with LLaVA-Med via a fusion decoder. The experiments demonstrate that PRS-Med achieves good segmentation and reasoning performance.

**Strengths:**

1. The introduction of MedPos includes spatially grounded QA pairs across six imaging modalities, it provides a valuable resource for research on explainable medical vision–language learning.

2. The integration of reasoning and segmentation sounds interesting. PRS-Med intends to combine spatial segmentation with natural language reasoning, with a focus on the gap between visual grounding and interpretability in medical AI.

3. Comprehensive evaluation results on both segmentation (Dice, IoU) and reasoning metrics (ROUGE, METEOR, accuracy) demonstrate a well-rounded analysis of model capability.

**Weaknesses:**

1. Lack of discussion and comparison with prior language-based prompt understanding and spatial reasoning works. Please see details in the Question section.

2. Limited novelty in dataset construction. The MedPos dataset generation process is not novel, and the dataset creation is not a challenging task either.

3. The motivation for generating textual spatial information is not well articulated: why is textual position reasoning needed when spatial coordinates can be directly derived from segmentation masks?

**Questions:**

1. Based on Figure 3, the model outputs both a mask and a textual description of the tumor’s location. However, since the mask already encodes precise spatial coordinates, what additional value does the textual position output provide? (The position can be computed directly from the mask). Could the authors clarify the motivation for generating textual spatial descriptions?

2. In Figure 1, the two-stage dataset construction pipeline appears very similar to Figure 3 in [1]. The dataset contribution seems limited, as this process could be easily replicated for existing medical imaging datasets that already include ground-truth masks: one could preprocess them once to obtain positional descriptions. Given the limited categories (top left, top right, etc), how significant is the benefit of this dataset in developing a generalizable positional reasoning model?
[1] FlanS: A Foundation Model for Free-Form Language-based Segmentation in Medical Images.

3. It is unclear whether PRS-Med and the baseline models are trained under the same conditions. Is PRS-Med trained on the newly constructed MedPos dataset, while the baselines are directly evaluated without additional fine-tuning? If so, does this create a potential fairness issue, since the test data likely shares distributional characteristics with MedPos training data, giving PRS-Med an advantage not available to the baselines? Could the authors clarify how baseline training and evaluation were conducted to ensure comparability?

---

### Official Review · Reviewer_x74u · 2025-11-01

**Soundness:** 2
**Presentation:** 1
**Contribution:** 2
**Rating:** 2
**Confidence:** 5

**Summary:**

This work present a vision-language model (VLM) for position reasoning segmentation for 2D medical images, which can answer questions by referring to the relevant visual locations. Specifically, authors first created a dedicated dataset with GPT-4 assisted multi-modal positional reasoning segmentation pipeline, followed by training a vision-language model with lora. The curated dataset and trained models will be released as a benchmark.

**Strengths:**

1. The paper is well-structured, and its logical flow is clear and easy to follow.

2. It is the first to introduce the concept of “location inference segmentation” in medical imaging, effectively incorporating spatial reasoning into the segmentation task. This idea is forward-looking and demonstrates strong potential for practical applications.

3. The work conducts comprehensive multi-modal evaluation across six imaging modalities, highlighting the approach’s broad applicability and generalizability.

**Weaknesses:**

**Limited Novelty**: The technical contribution is relatively limited. The method primarily integrates existing components rather than introducing substantial algorithmic innovation.

**Limited Query Diversity**: The query construction relies heavily on pre-defined templates, resulting in restricted instruction diversity. The model is likely overfitting to these simple prompts, which may undermine its language generalization capability. However, the paper does not provide experiments to evaluate this concern.

**Unclear Clinical Significance**: The clinical motivation and real-world value of the proposed “reasoning segmentation” task remain insufficiently justified in the medical context. The manuscript needs clearer explanation and concrete use cases to demonstrate the practical clinical relevance.

**Unfair Comparison and Evaluation Setup**: The proposed model is fine-tuned specifically for segmentation and spatial reasoning, whereas baseline models are general-purpose systems evaluated without similar task-specific adaptation. Although all methods are tested on the same task, only the proposed model receives fine-tuning, making it difficult to attribute performance gains to model design rather than privileged training. Additionally, all datasets used are publicly available, and no new benchmark is introduced. This raises concerns that prior methods—if fine-tuned on the same data—might surpass the proposed approach. Notably, BiomedParse already achieves comparable performance even without fine-tuning, further questioning the advantage claimed.

**Questions:**

1. Missing Comparison with Visual Grounding Models: Many existing large vision–language models are capable of visual grounding, and they could also be combined with segmentation modules to perform similar tasks. However, the paper does not include comparisons with such settings, which limits the completeness of the evaluation.

2. Insufficient Baseline Selection: Comparing only with LISA is inadequate and also unfair. The paper should include comparisons with recent medical MLLMs to provide a more convincing and comprehensive performance evaluation.

---

### Official Review · Reviewer_WUVy · 2025-11-01

**Soundness:** 3
**Presentation:** 2
**Contribution:** 2
**Rating:** 4
**Confidence:** 4

**Summary:**

The authors create a new dataset MedPos to address the lack of existing data on positional reasoning in medical images. The proposed PRS-Med model integrates a pre-trained multimodal LLM (adapted from LLaVA-Med via LoRA) with a lightweight TinySAM image encoder, fusing image features and text embeddings through a position-aware multi-head attention mechanism to produce a segmentation mask and a textual description of the tumor’s location.

**Strengths:**

The authors created the MedPos dataset generation pipeline to fill the data gap, using GPT-4 and expert validation to produce diverse, spatially-grounded Q&A pairs. This is a significant contribution that enables the task.
They evaluate PRS-Med on six different imaging modalities with standard segmentation metrics (mDice, mIoU) and language metrics (ROUGE, METEOR), and compare against strong baselines (SAM-Med2D, BiomedParse, LISA, LLaVA-Med, etc.).

**Weaknesses:**

1) While the MedPos dataset is creatively constructed, it relies on synthetic question templates generated by GPT-4 (albeit doctor-validated). This raises minor concerns about how closely the questions mirror genuine clinical inquiries.
2) The evaluation of “position understanding” accuracy is done using other LLMs as judges (Qwen 2.5 and Llama 3.1), it may introduce uncertainty or bias, since it depends on the evaluator LLMs correctly interpreting the model’s answer. Its reliability is not fully proven. A human evaluation of the spatial descriptions would strengthen the results.
3) The paper could be polished further in presentation. For example, Table 3 is labeled as showing “Qualitative results” when it actually reports quantitative accuracy percentages. There are a few minor grammatical errors and phrasing issues.
4) Additionally, a deeper discussion of failure cases or limitations is absent – acknowledging where PRS-Med might struggle (such as very small lesions or ambiguous questions) would provide a more balanced assessment of the work.

**Questions:**

1) please verify the how closely synthetic question templates mirror the real clinical problems.
2) please involve human evaluation instead of only using LLM as judge.
3) please proofread the whole paper.

---

### Note · Authors · 2025-11-28

**Comment:**

I want to withdraw this paper to improve it better

**Withdrawal Confirmation:**

I have read and agree with the venue's withdrawal policy on behalf of myself and my co-authors.